# Universal Approximation for Log-concave Distributions using Well-conditioned Normalizing Flows

Holden Lee [1]   Chirag Pabbaraju [2]   Anish Sevekari [2]   Andrej Risteski [2]

## Abstract

Affine-coupling models (Dinh et al., 2014; 2016) are a common type of normalizing flows, for which the Jacobian of the latent-to-observable-variable transformation is triangular, allowing the likelihood to be computed in linear time. Despite the widespread usage of affine couplings, the special structure of the architecture makes understanding their representational power challenging. The question of universal approximation was only recently resolved by three parallel papers (Huang et al., 2020; Zhang et al., 2020; Koehler et al., 2020) – who showed reasonably regular distributions can be approximated arbitrarily well using affine couplings – albeit with networks with a nearly-singular Jacobian. As ill-conditioned Jacobians are an obstacle for likelihood-based training, the fundamental question remains: which distributions can be approximated using *well-conditioned* affine coupling flows? In this paper, we show that any *log-concave* distribution can be approximated using well-conditioned affine-coupling flows. In terms of proof techniques, we uncover deep connections between affine coupling architectures, underdamped Langevin dynamics (a stochastic differential equation often used to sample from Gibbs measures) and Hénon maps (a structured dynamical system that appears in the study of symplectic diffeomorphisms). In terms of informing practice, we approximate a padded version of the input distribution with iid Gaussians – a strategy which (Koehler et al., 2020) empirically observed to result in better-conditioned flows, but had hitherto no theoretical grounding. Our proof can thus be seen as providing theoretical evidence for the benefits of Gaussian padding when training normalizing flows.

[1]Duke University [2]Carnegie Mellon University. Correspondence to: Andrej Risteski <aristesk@andrew.cmu.edu>.

Third workshop on *Invertible Neural Networks, Normalizing Flows, and Explicit Likelihood Models* (ICML 2021). Copyright 2021 by the author(s).

## 1. Introduction

Normalizing flows (Dinh et al., 2014; Rezende & Mohamed, 2015) are a class of generative models parametrizing a distribution in $\mathbb{R}^d$ as the pushfoward of a simple distribution (e.g. Gaussian) through an invertible map $g_\theta : \mathbb{R}^d \to \mathbb{R}^d$ with trainable parameter $\theta$. The fact that $g_\theta$ is invertible allows us to write down an explicit expression for the density of a point $x$ through the change-of-variables formula, namely $p_\theta(x) = \phi(g_\theta^{-1}(x))\det(Dg_\theta^{-1}(x))$, where $\phi$ denotes the density of the standard Gaussian. For different choices of parametric families for $g_\theta$, one gets different families of normalizing flows, e.g. affine coupling flows (Dinh et al., 2014; 2016; Kingma & Dhariwal, 2018), Gaussianization flows (Meng et al., 2020), sum-of-squares polynomial flows (Jaini et al., 2019).

In this paper we focus on affine coupling flows – arguably the family that has been most successfully scaled up to high resolution datasets (Kingma & Dhariwal, 2018). The parametrization of $g_\theta$ is chosen to be a composition of so-called *affine coupling blocks*, which are maps $f : \mathbb{R}^d \to \mathbb{R}^d$, s.t. $f(x_S, x_{[d]\setminus S}) = (x_S, x_{[d]\setminus S} \odot s(x_S) + t(x_S))$, where $\odot$ denotes entrywise multiplication and $s, t$ are (typically simple) neural networks. The choice of parametrization is motivated by the fact that the Jacobian of each affine block is triangular, so that the determinant can be calculated in linear time.

Despite the empirical success of this architecture, theoretical understanding remains elusive. The most basic questions revolve around the representational power of such models. Even the question of universal approximation was only recently answered by three concurrent papers (Huang et al., 2020; Zhang et al., 2020; Koehler et al., 2020)—though in a less-than-satisfactory manner, in light of how normalizing flows are trained. Namely, (Huang et al., 2020; Zhang et al., 2020) show that any (reasonably well-behaved) distribution $p$, once padded with zeros and treated as a distribution in $\mathbb{R}^{d+d'}$, can be arbitrarily closely approximated by an affine coupling flow. While such padding can be operationalized as an algorithm by padding the training image with zeros, it is never done in practice, as it results in an ill-conditioned Jacobian. This is expected, as the map that always sends the last $d'$ coordinates to 0 is not injective. (Koehler et al., 2020)

prove universal approximation without padding; however their construction *also* gives rise to a poorly conditioned Jacobian: namely, to approximate a distribution $p$ to within accuracy $\epsilon$ in the Wasserstein-1 distance, the Jacobian of the network they construct will have smallest singular value on the order of $\epsilon$.

Importantly, for all these constructions, the condition number of the resulting affine coupling map is poor *no matter how nice the underlying distribution it's trying to approximate is*. In other words, the source of this phenomenon isn't that the underlying distribution is low-dimensional or otherwise degenerate. Thus the question arises:

**Question:** *Can well-behaved distributions be approximated by an affine coupling flow with a well-conditioned Jacobian?*

In this paper, we answer the above question in the affirmative for a broad class of distributions – log-concave distributions – if we pad the input distribution not with zeroes, but with independent Gaussians. This gives theoretical grounding of an empirical observation in (Koehler et al., 2020) that Gaussian padding works better than zero-padding, as well as no padding.

The practical relevance of this question is in providing guidance on the type of distributions we can hope to fit via training using an affine coupling flow. Theoretically, our techniques uncover some deep connections between affine coupling flows and two other (seeming unrelated) areas of mathematics: *stochastic differential equations* (more precisely *underdamped Langevin dynamics*, a "momentum" variant of the standard overdamped Langevin dynamics) and *dynamical systems* (more precisely, a family of dynamical systems called *Hénon-like maps*).

## 2. Overview of results

In order to state our main result, we introduce some notation and definitions.

### 2.1. Notation

**Definition 1.** An *affine coupling block* is a map $f : \mathbb{R}^d \to \mathbb{R}^d$, s.t. $f(x_S, x_{[d]\setminus S}) = (x_S, x_{[d]\setminus S} \odot s(x_S) + t(x_S))$ for some set of coordinates $S$, where $\odot$ denotes entrywise multiplication and $s, t$ are trainable (generally non-linear) functions. An *affine coupling network* is a finite sequence of affine coupling blocks. Note that the partition $(S, [d] \setminus S)$, as well as $s, t$ may be different between blocks. We say that the non-linearities are in a class $\mathcal{F}$ (e.g., neural networks, polynomials, etc.) if $s, t \in \mathcal{F}$.

The appeal of affine coupling networks comes from the fact that the Jacobian of each affine block is triangular, so calculating the determinant is a linear-time operation.

We will be interested in the *conditioning* of $f$—that is, an upper bound on the largest singular value $\sigma_{\max}(Df)$ and lower bound on the smallest singular value $\sigma_{\min}(Df)$ of the Jacobian $Df$ of $f$. Note that this is a slight abuse of nomenclature – most of the time, "condition number" refers to the ratio of the largest and smallest singular value. As training a normalizing flow involves evaluating $\det(Df)$, we in fact want to ensure that neither the smallest nor largest singular values are extreme.

The class of distributions we will focus on approximating via affine coupling flows is *log-concave* distributions:

**Definition 2.** A distribution $p : \mathbb{R}^d \to \mathbb{R}^+, p(x) \propto e^{-U(x)}$ is *log-concave* if $\nabla^2 U(x) = -\nabla^2 \ln p(x) \succeq 0$.

Log-concave distributions are typically used to model distributions with Gaussian-like tail behavior. What we will leverage about this class of distributions is that a special stochastic differential equation (SDE), called *underdamped Langevin dynamics*, is well-behaved in an analytic sense. Finally, we recall the definitions of positive definite matrices and Wasserstein distance, and introduce a notation for truncated distributions.

**Definition 3.** We say that a symmetric matrix is *positive semidefinite (PSD)* if all of its eigenvalues are non-negative. For symmetric matrices $A, B$, we write $A \succeq B$ if and only if $A - B$ is PSD.

**Definition 4.** Given two probability measures $\mu, \nu$ over a metric space $(M, d)$, the *Wasserstein-1 distance* between them, denoted $W_1(\mu, \nu)$, is defined as

$$W_1(\mu, \nu) = \inf_{\gamma \in \Gamma(\mu,\nu)} \int_{M \times M} d(x, y)\, d\gamma(x, y)$$

where $\Gamma(\mu, \nu)$ is the set of couplings, i.e. measures on $M \times M$ with marginals $\mu, \nu$ respectively. For two probability *distributions* $p, q$, we denote by $W_1(p, q)$ the Wasserstein-1 distance between their associated measures. In this paper, we set $M = \mathbb{R}^d$ and $d(x, y) = \|x - y\|_2$.

**Definition 5.** Given a distribution $q$ and a compact set $\mathcal{C}$, we denote by $q|_{\mathcal{C}}$ the distribution $q$ truncated to the set $\mathcal{C}$. The truncated measure is defined as $q|_{\mathcal{C}}(A) = \frac{1}{q(\mathcal{C})} q(A \cap \mathcal{C})$.

### 2.2. Main result

Our main result states that we can approximate any log-concave distribution in Wasserstein-1 distance by a *well-conditioned* affine-coupling flow network. Precisely, we show:

**Theorem 1.** *Let $p(x) : \mathbb{R}^d \to \mathbb{R}^+$ be of the form $p(x) \propto e^{-U(x)}$, such that:*

1. *$U \in C^2$, i.e., $\nabla^2 U(x)$ exists and is continuous.*
2. *$\ln p$ satisfies $\mathrm{I}_d \preceq -\nabla^2 \ln p(x) \preceq \kappa \mathrm{I}_d$.*

*Furthermore, let $p_0 := p \times \mathcal{N}(0, I_d)$. Then, for every $\epsilon > 0$, there exists a compact set $\mathcal{C} \subset \mathbb{R}^{2d}$ and an invertible affine-coupling network $f : \mathbb{R}^{2d} \to \mathbb{R}^{2d}$ with polynomial non-linearities, such that*

$$W_1(f_{\#}(\mathcal{N}(0, I_{2d})|_{\mathcal{C}}), p_0) \leq \epsilon.$$

*Furthermore, the map defined by this affine-coupling network $f$ is well conditioned over $\mathcal{C}$, that is, there are positive constants $A(\kappa), B(\kappa) = \kappa^{O(1)}$ such that for any unit vector $w$,*

$$A(\kappa) \leq \|D_w f(x, v)\| \leq B(\kappa)$$

*for all $(x, v) \in \mathcal{C}$, where $D_w$ is the directional derivative in the direction $w$. In particular, the condition number of $Df(x, v)$ is bounded by $\frac{B(\kappa)}{A(\kappa)} = \kappa^{O(1)}$ for all $(x, v) \in \mathcal{C}$.*

We make several remarks regarding the statement of the theorem:

*Remark* 1. The Gaussian padding (i.e. setting $p_0 = p \times \mathcal{N}(0, I_d)$) is essential for our proofs. All the other prior works on the universal approximation properties of normalizing flows (with or without padding) result in ill-conditioned affine coupling networks. This gives theoretical backing of empirical observations on the benefits of Gaussian padding in (Koehler et al., 2020).

*Remark* 2. The choice of non-linearities $s, t$ being polynomials is for the sake of convenience in our proofs. Using standard universal approximation results, they can also be chosen to be neural networks with a smooth activation function.

*Remark* 3. The Jacobian $Df$ has both upper-bounded largest singular value, and lower-bounded smallest singular value—which of course bounds the determinant $\det(Df)$. As remarked in Section 2.1, merely bounding the ratio of the two quantities would not suffice for this. Moreover, the bound we prove *only* depends on properties of the distribution (i.e., $\kappa$), and does not worsen as $\epsilon \to 0$, in contrast to (Koehler et al., 2020).

*Remark* 4. The region $\mathcal{C}$ where the pushforward of the Gaussian through $f$ and $p_0$ are close is introduced solely for technical reasons—essentially, standard results in analysis for approximating smooth functions by polynomials can only be used if the approximation needs to hold on a compact set. Note that $\mathcal{C}$ can be made arbitrarily large by making $\epsilon$ arbitrarily small.

*Remark* 5. We do not provide a bound on the number of affine coupling blocks, although a bound can be extracted from our proofs.

## 3. Proof Sketch of Theorem 1

We wish to construct an affine coupling network that (approximately) pushes forward a Gaussian $p^* = \mathcal{N}(0, I_{2d})$ to

the distribution we wish to model with Gaussian padding, i.e. $p_0 = p \times \mathcal{N}(0, I_d)$. Because the inverse of an affine coupling network is an affine coupling network, we can invert the problem, and instead attempt to map $p_0$ to $N(0, I_{2d})$. [1]

There is a natural map that takes $p_0$ to $p^* = N(0, I_{2d})$, namely, underdamped Langevin dynamics (1). Hence, our proof strategy involves understanding and simulating underdamped Langevin dynamics with the initial distribution $p_0 = p \times \mathcal{N}(0, I_d)$, and the target distribution $p^* = \mathcal{N}(0, I_{2d})$, and comprises of two important steps.

First, we show that the flow-map for underdamped Langevin is well-conditioned. Here, by flow-map, we mean the map which assigns each $x$ to its evolution over a certain amount of time $t$ according to the equations specified by (1):

$$\begin{cases} dx_t &= -\zeta v_t dt \\ dv_t &= -\gamma \zeta v_t dt - \nabla U(x_t) dt + \sqrt{2\gamma} \, dB_t. \end{cases} \quad (1)$$

The stationary distribution of the SDEs (limiting distribution as $t \to \infty$) is given by $p^*(x, v) \propto e^{-U(x) - \frac{\zeta}{2}\|v\|^2}$.

The convergence of (1) can be bounded when the distribution $p(x) \propto \exp(-U(x))$ satisfies an analytic condition, namely has a bounded *log-Sobolev* constant. For brevity, we omit the definition of a log-Sobolev inequality, since we will only need the following fact:

**Fact 1** ((Bakry & Émery, 1985; Bakry et al., 2013))**.** Let the distributions $p(x) \propto \exp(-U(x))$ be such that $U(x) \succeq \lambda I$. Then, $p$ has log-Sobolev constant bounded by $\lambda$.

In fact, our proofs leverage a less well-known *deterministic* form of the updates which is equivalent to (1). Precisely, we convert (1) to an equivalent ODE (with time-dependent coefficients). The proof of this fact (via a straightforward comparison of the Fokker-Planck equation) can be found in (Ma et al., 2019).

**Theorem 2.** *Let $p_t(x_t, v_t)$ be the probability distribution of running (1) for time $t$. If started from $(x_0, v_0) \sim p_0$, the probability distribution of the solution $(x_t, v_t)$ to the ODEs*

$$\frac{d}{dt} \begin{bmatrix} x_t \\ v_t \end{bmatrix} = \begin{bmatrix} O & I_d \\ -I_d & -\gamma I_d \end{bmatrix} (\nabla \ln p_t - \nabla \ln p^*) \quad (2)$$

*is also $p_t(x_t, v_t)$.*

With this in mind, we show the following fact about the conditioning of the underdamped Langevin flow:

**Lemma 1.** *Consider underdamped Langevin dynamics (1) with $\zeta = 1$, friction coefficient $\gamma < 2$ and starting distribution $p$ which satisfies all the assumptions in Theorem 1. Let $T_t$ denote the flow map from time 0 to time $t$ induced by (1).*

---

[1] As an aside, a similar strategy is taken in practice by recent SDE-based generative models ((Song et al., 2020)).

*Then for any $x_0, v_0 \in \mathbb{R}^d$ and unit vector $w$, the directional derivative of $T_t$ at $x_0, v_0$ in direction $w$ satisfies*

$$\|D_w T_t(x_0)\| \geq \left(1 + \frac{2+\gamma}{2-\gamma}(\kappa - 1)\right)^{-2/\gamma},$$

$$\|D_w T_t(x_0)\| \leq \left(1 + \frac{2+\gamma}{2-\gamma}(\kappa - 1)\right)^{2/\gamma}.$$

*Therefore, the condition number of $T_t$ is bounded by $\left(1 + \frac{2+\gamma}{2-\gamma}(\kappa - 1)\right)^{4/\gamma}$.*

The main idea is to consider how $\nabla^2 \ln p_t$ evolves if we replace (1) by its discretization,

$$\widetilde{x}_{t+\eta} = \widetilde{x}_t + \eta \widetilde{v}_t$$
$$\widetilde{v}_{t+\eta} = (1 - \eta\gamma)\widetilde{v}_t - \eta\widetilde{x}_t + \xi_t, \quad \xi_t \sim N(0, 2\eta I_d).$$

Because the stationary distribution is a Gaussian, $\nabla U(x_t) = x_t$ in (1) and the above equations take a particularly simple form: we apply a linear transformation to $\begin{bmatrix} \widetilde{x}_t \\ \widetilde{v}_t \end{bmatrix}$, and then add Gaussian noise, which corresponds to convolving the current distribution by a Gaussian. The core of the approach is then to track upper and lower bounds for $\nabla^2 \ln p_t$, and compute how they evolve under this linear transformation and convolution by a Gaussian.

Second, we break the simulation of underdamped Langevin dynamics for a certain time $t$ into intervals of size $\tau$, and show that the *inverse* flow-map over each $\tau$-sized interval of time can be approximated well by a composition of affine-coupling maps. To show this, we consider a more general system of ODEs than the one in (Turaev, 2002) (in particular, a non-Hamiltonian system), which can be applied to *underdamped* Langevin dynamics:

$$\begin{cases} \frac{dx}{dt} = \frac{\partial}{\partial v} H(x, v, t) \\ \frac{dv}{dt} = -\frac{\partial}{\partial x} H(x, v, t) - \gamma \frac{\partial}{\partial v} H(x, v, t) \end{cases} \quad (3)$$

In (Turaev, 2002), the version of the above system where $\gamma = 0$ was proven to be a *universal approximator* in some sense: namely, the iterations of this ODE can approximate any *symplectic diffeomorphism*: a continuous map which preserves volumes (i.e. the Jacobian of the map is 1). These kinds of diffeomorphisms have their genesis in Hamiltonian formulations of classical mechanics (Abraham & Marsden, 2008). Precisely, after reducing the problem to considering polynomial $H$ only, we show:

**Lemma 2.** *Let $\mathcal{C} \subset \mathbb{R}^n$ be a compact set. For any function $H(x, v, t) : \mathbb{R}^{2d} \to \mathbb{R}$ which is polynomial in $(x, v)$, there exist polynomial functions $J, F, G$, s.t. the time-$(t_0, t_0 + \tau)$ flow map of the system*

$$\begin{cases} \frac{dx}{dt} = \frac{\partial}{\partial v} H(x, v, t) \\ \frac{dv}{dt} = -\frac{\partial}{\partial x} H(x, v, t) - \gamma \frac{\partial}{\partial v} H(x, v, t) \end{cases} \quad (4)$$

*is uniformly $O(\tau^2)$-close in $C^1$ over $\mathcal{C}$ topology to the time-$2\pi$ map of the system*

$$\begin{cases} \frac{dx}{dt} = v - \tau F(v, t) \odot x \\ \frac{dv_j}{dt} = -\Omega_j^2 x_j - \tau J_j(x, t) - \tau v_j G_j(x, t) \end{cases} \quad (5)$$

*Here, $\odot$ denotes component-wise product, and the constants inside the $O(\cdot)$ depend on $\mathcal{C}$.*

We then show that the *inverse* flow-map of this system of ODEs can be approximated by a sequence of affine-coupling blocks, by considering an Euler discretization of the newly constructed ODE (5) into small steps of size $\eta$ i.e.

$$\begin{cases} x_{n+1} = x_n + \eta(v_n - \tau F(v_n, \eta n) \odot x_n) \\ v_{n+1,j} = v_{n,j} - \eta(\Omega_j^2 x_{n,j} - \tau J_j(x_n, \eta n) \\ \qquad\qquad - \tau v_{n,j} G_j(x_n, \eta n)) \end{cases} \quad (6)$$

Note that each step above can be written as a composition of affine coupling blocks given by

$$(x_n, v_n) \mapsto (x_n, v_{n+1}) \mapsto (x_{n+1}, v_{n+1})$$

## 4. Related Work

On the theory front for flow models, most closely related to our results are the recent works of (Huang et al., 2020; Zhang et al., 2020; Koehler et al., 2020). The former two show universal approximation of affine couplings – albeit if the input is padded with zeros. This of course results in maps with singular Jacobians, which is why this strategy isn't used in practice. (Koehler et al., 2020) show universal approximation without padding – though their constructions results in a flow model with condition number $1/\epsilon$ to get approximation $\epsilon$ in the Wasserstein sense, regardless of how well-behaved the distribution to be approximated is. Furthemore, (Koehler et al., 2020) provide some empirical evidence that padding with iid Gaussians (as in our paper) is better than both zero padding (as in (Huang et al., 2020; Zhang et al., 2020)) and no padding on small-scale data.

## 5. Conclusion

In this paper, we provide the first guarantees on universal approximation with *well-conditioned* affine coupling networks. The conditioning of the network is crucial in order for likelihood-based training to succeed. At the mathematical level, we uncover connections between stochastic differential equations, dynamical systems and affine coupling flows. Our construction uses Gaussian padding, which lends support to the empirical observation that this strategy tends to result in better-conditioned flows (Koehler et al., 2020). We leave it as an open problem to generalize beyond log-concave distributions.

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

# A. Appendix

For a complete version of this paper that includes proofs of all main results and supporting technical details, please refer to the arXiv version at https://arxiv.org/abs/2107.02951.