# OpenReview forum: "Universal Approximation for Log-concave Distributions using Well-conditioned Normalizing Flows"
_ICML.cc/2021/Workshop/INNF — INNF+ 2021 spotlighttalk_

### Official Review · Reviewer_z695 · 2021-06-07

**Rating:** Accept
**Confidence:** 4

**Summary:**

The authors prove bipartite flows can approximate any log-concave distribution while maintaining
upper and lower bounds on the eigenvalues of the Jacobians (of transformations that define the flow).
The results rely on padding the target random variable with Gaussian noise. The proof has two steps:
i) the authors show an underdamped Langevin dynamic which maps between the flow prior and the target; and
ii) approximate the Langevin dynamic up to a time $t$ using a discrete number of affine couplings.

**Justification For Rating:**

This is a good contribution. The proof technique is interesting and the paper is relatively well written.

I would have a different title, since readers may expect more expressiveness when they see 'universal'.
In fact, I would say log-concavity is very restrictive and I advise authors to be cautious about it (line 72: 'broad class of distributions' might be too strong).

I also think the manuscript would benefit from a discussion about assumption 2 (listed) in Theorem 1. Which densities obey this and which do not? Maybe a remark would be good.

Besides this, I would appreciate a small overview of the [structure of the] proof at the beginning of Section 3.

I am looking forward to seeing an extended version in the future.

---

### Official Review · Reviewer_uqgP · 2021-06-11

**Rating:** Accept
**Confidence:** 3

**Summary:**

Authors answer the question: "Can well-behaved distributions be approximated by an afﬁne coupling ﬂow with a well-conditioned Jacobian?" They answer positively, proving that log-concave distributions belong the the class of distributions that can be approximated by such flows, if the input distribution is padded with independent Gaussians.


**Justification For Rating:**

# Strengths
The main strength of this work is its theoretical soundness and the fact that it overcomes the limitations of previous work, namely the ill-conditioned Jacobian yielded by their assumptions.
Additionally, the independent Gaussian padding assumption, backs the previous empirical results showing their usefulness to get better modelling capacity.
The bound only depends on properties of the distribution (and does not worsen as epsilon gets small, as opposed to Koehler et al., 2020), which is satisfying.

# Weaknesses
- How broad the class of log-concave distribution is?
- How hard is it to describe the entire space of distributions that can be "approximated by an afﬁne coupling ﬂow with a well-conditioned Jacobian"?

# Correctness / Clarity / Relation to prior work
The proof is divided in three parts:
1/ The flow generated by specific underdamped Langevin dynamics yields a limiting distribution that is the distribution of interest (up to some Gaussian noise)
2/ Such flow is well-conditioned
3/ Intervals of the (inverse) underdamped Langevin dynamics can be approximated by a composition of affine-coupling maps.

If I understood correctly, the affine-coupling split is between x and v, representing respectively a position and velocity. Hence, this velocity is associated with the "padded" dimensions and can be regarded as an augmented / auxiliary variable. The Gaussian padding is therefore a Gaussian prior for the velocity. Perhaps it would be useful to stress this perspective, as it could help understanding both the proof and the motivation of the padding (and an optimal choice of split?) for readers with a background in methods like HMC.

Otherwise the writing is fairly good. Section 3 could be better organised to help the reader progressing into the proof. I am familiar with the mathematical concepts used, yet not no expert so I cannot guarantee the correctness of the proof. It does appear to be sound.

# Additional feedback
- l 026: It would perhaps be useful to explain the reasons why "ill-conditioned Jacobians are an obstacle for likelihood-based training"
- l 030: Is this strict? As in "well-conditioned afﬁne-coupling ﬂows can *only* model log-concave distribution"?
- l 075 (right): space missing
- Theorem 1: Would be useful for clarity to name the assumptions (e.g. log-concavity, iid Gaussian padding)
- Theorem 1: Is the padding Gaussian because the base distribution is Gaussian? What is the intuition for the need of the padding?
- l 196: "isn’t"

---

### Decision · Program_Chairs · 2021-06-15

**Decision:**

Accept (spotlight talk)

**Comment:**

The topic of this paper fits well in the workshop and the reviews were positive. Therefore we have accepted this paper with a spotlight talk. For the camera ready paper, please take into account the suggestion by one of the reviewers to be more cautious with claims on universal approximations by this model class if it can only approximate log-concave distributions.